# Learning Neural PDE Solvers with Convergence Guarantees

**Jun-Ting Hsieh***
Stanford
junting@stanford.edu

**Shengjia Zhao***
Stanford
sjzhao@stanford.edu

**Stephan Eismann**
Stanford
seismann@stanford.edu

**Lucia Mirabella**
Siemens
lucia.mirabella@siemens.com

**Stefano Ermon**
Stanford
ermon@stanford.edu

## Abstract

Partial differential equations (PDEs) are widely used across the physical and computational sciences. Decades of research and engineering went into designing fast iterative solution methods. Existing solvers are general purpose, but may be sub-optimal for specific classes of problems. In contrast to existing hand-crafted solutions, we propose an approach to learn a fast iterative solver tailored to a specific domain. We achieve this goal by learning to modify the updates of an existing solver using a deep neural network. Crucially, our approach is proven to preserve strong correctness and convergence guarantees. After training on a single geometry, our model generalizes to a wide variety of geometries and boundary conditions, and achieves 2-3 times speedup compared to state-of-the-art solvers.

## 1 Introduction

Partial differential equations (PDEs) are ubiquitous tools for modeling physical phenomena, such as heat, electrostatics, and quantum mechanics. Traditionally, PDEs are solved with hand-crafted approaches that iteratively update and improve a candidate solution until convergence. Decades of research and engineering went into designing update rules with fast convergence properties.

The performance of existing solvers varies greatly across application domains, with no method uniformly dominating the others. Generic solvers are typically effective, but could be far from optimal for specific domains. In addition, high performing update rules could be too complex to design by hand. In recent years, we have seen that for many classical problems, complex updates *learned* from data or experience can out-perform hand-crafted ones. For example, for Markov chain Monte Carlo, learned proposal distributions lead to orders of magnitude speedups compared to hand-designed ones (Song et al., 2017; Levy et al., 2017). Other domains that benefited significantly include learned optimizers (Andrychowicz et al., 2016) and learned data structures (Kraska et al., 2018). Our goal is to bring similar benefits to PDE solvers.

Hand-designed solvers are relatively simple to analyze and are guaranteed to be correct in a large class of problems. The main challenge is how to provide the same guarantees with a potentially much more complex learned solver. To achieve this goal, we build our learned iterator on top of an existing standard iterative solver to inherit its desirable properties. The iterative solver updates the solution at each step, and we learn a parameterized function to modify this update. This function class is chosen so that for any choice of parameters, the fixed point of the original iterator is preserved. This guarantees correctness, and training can be performed to enhance convergence speed. Because of this design, we only train on a single problem instance; our model correctly generalizes to a variety of different geometries and boundary conditions with no observable loss of performance. As a result, our approach provides: (i) theoretical guarantees of convergence to the correct stationary solution, (ii) faster convergence than existing solvers, and (iii) generalizes to geometries and boundary conditions very different from the ones seen at training time. This is in stark contrast with

existing deep learning approaches for PDE solving (Tang et al., 2017; Farimani et al., 2017) that are limited to specific geometries and boundary conditions, and offer no guarantee of correctness.

Our approach applies to any PDE with existing linear iterative solvers. As an example application, we solve the 2D Poisson equations. Our method achieves a 2-3× speedup on number of multiply-add operations when compared to standard iterative solvers, even on domains that are significantly different from our training set. Moreover, compared with state-of-the-art solvers implemented in FEniCS (Logg et al., 2012), our method achieves faster performance in terms of wall clock CPU time. Our method is also simple as opposed to deeply optimized solvers such as our baseline in FEniCS (minimal residual method + algebraic multigrid preconditioner). Finally, since we utilize standard convolutional networks which can be easily parallelized on GPU, our approach leads to an additional 30× speedup when run on GPU.

## 2 BACKGROUND

In this section, we give a brief introduction of linear PDEs and iterative solvers. We refer readers to LeVeque (2007) for a thorough review.

### 2.1 LINEAR PDEs

Linear PDE solvers find functions that satisfy a (possibly infinite) set of linear differential equations. More formally, let $\mathcal{F} = \{u : \mathbb{R}^k \to \mathbb{R}\}$ be the space of candidate functions, and $\mathcal{A} : \mathcal{F} \to \mathcal{F}$ be a linear operator; the goal is to find a function $u \in \mathcal{F}$ that satisfies a linear equation $\mathcal{A}u = f$, where $f$ is another function $\mathbb{R}^k \to \mathbb{R}$ given by our problem. Many PDEs fall into this framework. For example, heat diffusion satisfies $\nabla^2 u = f$ (Poisson equation), where $\nabla^2 = \frac{\partial^2}{\partial x_1^2} + \cdots + \frac{\partial^2}{\partial x_k^2}$ is the linear Laplace operator; $u$ maps spatial coordinates (e.g. in $\mathbb{R}^3$) into its temperature, and $f$ maps spatial coordinates into the heat in/out flow. Solving this equation lets us know the stationary temperature given specified heat in/out flow.

Usually the equation $\mathcal{A}u = f$ does not uniquely determine $u$. For example, $u = \text{constant}$ for any constant is a solution to the equation $\nabla^2 u = 0$. To ensure a unique solution we provide additional equations, called "boundary conditions". Several boundary conditions arise very naturally in physical problems. A very common one is the Dirichlet boundary condition, where we pick some subset $\mathcal{G} \subset \mathbb{R}^k$ and fix the values of the function on $\mathcal{G}$ to some fixed value $b$,

$$u(x) = b(x), \text{ for all } x \in \mathcal{G}$$

where the function $b$ is usually clear from the underlying physical problem. As in previous literature, we refer to $\mathcal{G}$ as the *geometry* of the problem, and $b$ as the *boundary value*. We refer to the pair $(\mathcal{G}, b)$ as the *boundary condition*. In this paper, we only consider linear PDEs and boundary conditions that have unique solutions.

### 2.2 FINITE DIFFERENCE METHOD

Most real-world PDEs do not admit an analytic solution and must be solved numerically. The first step is to discretize the solution space $\mathcal{F}$ from $\mathbb{R}^k \to \mathbb{R}$ into $\mathbb{D}^k \to \mathbb{R}$, where $\mathbb{D}$ is a discrete subset of $\mathbb{R}$. When the space is compact, it is discretized into an $n \times n \times n \cdots$ ($k$ many) uniform Cartesian grid with mesh width $h$. Any function in $\mathcal{F}$ is approximated by its value on the $n^k$ grid points. We denote the discretized function as a vector $u$ in $\mathbb{R}^{n^k}$. In this paper, we focus on 2D problems ($k = 2$), but the strategy applies to any dimension.

We discretize all three terms in the equation $\mathcal{A}u = f$ and boundary condition $(\mathcal{G}, b)$. The PDE solution $u$ is discretized such that $u_{i,j} = u(x_i, y_j)$ corresponds to the value of $u$ at grid point $(x_i, y_j)$. We can similarly discretize $f$ and $b$. In linear PDEs, the linear operator $\mathcal{A}$ is a linear combination of partial derivative operators. For example, for the Poisson equation $\mathcal{A} = \nabla^2 = \sum_i \frac{\partial^2}{\partial x_i^2}$. Therefore we can first discretize each partial derivative, then linearly combine the discretized partial derivatives to obtain a discretized $\mathcal{A}$.

Finite difference is a method that approximates partial derivatives in a discretized space, and as mesh width $h \to 0$, the approximation approaches the true derivative. For example, $\frac{\partial^2}{\partial x^2} u$ can

be discretized in 2D as $\frac{\partial^2}{\partial x^2} u \approx \frac{1}{h^2}(u_{i-1,j} - 2u_{i,j} + u_{i+1,j})$, the Laplace operator in 2D can be correspondingly approximated as:

$$\nabla^2 u = \frac{\partial^2 u}{\partial x^2} + \frac{\partial^2 u}{\partial y^2} \approx \frac{1}{h^2}(u_{i-1,j} + u_{i+1,j} + u_{i,j-1} + u_{i,j+1} - 4u_{i,j}) \tag{1}$$

After discretization, we can rewrite $\mathcal{A}u = f$ as a linear matrix equation

$$Au = f \tag{2}$$

where $u, f \in \mathbb{R}^{n^2}$, and $A$ is a matrix in $\mathbb{R}^{n^2 \times n^2}$ (these are $n^2$ dimensional because we focus on 2D problems). In many PDEs such as the Poisson and Helmholtz equation, $A$ is sparse, banded, and symmetric.

## 2.3 BOUNDARY CONDITION

We also need to include the boundary condition $u(x) = b(x)$ for all $x \in \mathcal{G}$. If a discretized point $(x_i, y_j)$ belongs to $\mathcal{G}$, we need to fix the value of $u_{i,j}$ to $b_{i,j}$. To achieve this, we first define $e \in \{0, 1\}^{n^2}$ to be a vector of 0's and 1's, in which 0 indicates that the corresponding point belongs to $\mathcal{G}$. Then, we define a "reset" matrix $G = diag(e)$, a diagonal matrix $\mathbb{R}^{n^2} \to \mathbb{R}^{n^2}$ such that

$$(Gu)_{i,j} = \begin{cases} u_{i,j} & (x_i, y_j) \notin \mathcal{G} \\ 0 & (x_i, y_j) \in \mathcal{G} \end{cases} \tag{3}$$

Intuitively $G$ "masks" every point in $\mathcal{G}$ to 0. Similarly, $I - G$ can mask every point not in $\mathcal{G}$ to 0. Note that the boundary values are fixed and do not need to satisfy $Au = f$. Thus, the solution $u$ to the PDE under geometry $\mathcal{G}$ should satisfy:

$$\begin{aligned} G(Au) &= Gf \\ (I - G)u &= (I - G)b \end{aligned} \tag{4}$$

The first equation ensures that the interior points (points not in $\mathcal{G}$) satisfy $Au = f$, and the second ensures that the boundary condition is satisfied.

To summarize, $(\mathcal{A}, \mathcal{G}, f, b, n)$ is our *PDE problem*, and we first discretize the problem on an $n \times n$ grid to obtain $(A, G, f, b, n)$. Our objective is to obtain a solution $u$ that satisfies Eq. (4), i.e. $Au = f$ for the interior points and boundary condition $u_{i,j} = b_{i,j}$, $\forall(x_i, y_j) \in \mathcal{G}$.

## 2.4 ITERATIVE SOLVERS

A *linear* iterative solver is defined as a function that inputs the current proposed solution $u \in \mathbb{R}^{n^2}$ and outputs an updated solution $u'$. Formally it is a function $\Psi : \mathbb{R}^{n^2} \to \mathbb{R}^{n^2}$ that can be expressed as

$$u' = \Psi(u) = Tu + c \tag{5}$$

where $T$ is a constant update matrix and $c$ is a constant vector. For each iterator $\Psi$ there may be special vectors $u^* \in \mathbb{R}^{n^2}$ that satisfy $u^* = \Psi(u^*)$. These vectors are called fixed points.

The iterative solver $\Psi$ should map any initial $u^0 \in \mathbb{R}^{n^2}$ to a correct solution of the PDE problem. This is formalized in the following theorem.

**Definition 1** (Valid Iterator). *An iterator $\Psi$ is valid w.r.t. a PDE problem $(A, G, f, b, n)$ if it satisfies:*

    a) ***Convergence:*** *There is a unique fixed point $u^*$ such that $\Psi$ converges to $u^*$ from any initialization: $\forall u^0 \in \mathbb{R}^{n^2}, \lim_{k\to\infty} \Psi^k(u^0) = u^*$.*

    b) ***Fixed Point:*** *The fixed point $u^*$ is the solution to the linear system $Au = f$ under boundary condition $(G, b)$.*

**Convergence:** Condition (a) in Definition 1 is satisfied if the matrix $T$ is *convergent*, i.e. $T^k \to 0$ as $k \to \infty$. It has been proven that $T$ is convergent if and only if the spectral radius $\rho(T) < 1$ (Olver, 2008):

**Theorem 1.** *(Olver, 2008, Prop 7.25) For a linear iterator $\Psi(u) = Tu + c$, $\Psi$ converges to a unique stable fixed point from any initialization if and only if the spectral radius $\rho(T) < 1$.*

*Proof.* See Appendix A. □

It is important to note that Condition (a) only depends on $T$ and not the constant $c$.

**Fixed Point:** Condition (b) in Definition 1 contains two requirements: satisfy $Au = f$, and the boundary condition $(G, b)$. To satisfy $Au = f$ a standard approach is to design $\Psi$ by matrix splitting: split the matrix $A$ into $A = M - N$; rewrite $Au = f$ as $Mu = Nu + f$ (LeVeque, 2007). This naturally suggests the iterative update

$$u' = M^{-1}Nu + M^{-1}f \tag{6}$$

Because Eq. (6) is a rewrite of $Au = f$, stationary points $u^*$ of Eq. (6) satisfy $Au^* = f$. Clearly, the choices of $M$ and $N$ are arbitrary but crucial. From Theorem 1, we must choose $M$ such that the update converges. In addition, $M^{-1}$ must easy to compute (e.g., diagonal).

Finally we also need to satisfy the boundary condition $(I - G)u = (I - G)b$ in Eq.4. After each update in Eq. (6), the boundary condition could be violated. We use the "reset" operator defined in Eq. (3) to "reset" the values of $u_{i,j}$ to $b_{i,j}$ by $Gu + (I - G)b$.

The final update rule becomes

$$u' = G(M^{-1}Nu + M^{-1}f) + (I - G)b \tag{7}$$

Despite the added complexity, it is still a linear update rule in the form of $u' = Tu + c$ in Eq. (5): we have $T = GM^{-1}N$ and $c = GM^{-1}f + (1 - G)b$. As long as $M$ is a full rank diagonal matrix, fixed points of this equation satisfies Eq. (4). In other words, such a fixed point is a solution of the PDE problem $(A, G, f, b, n)$.

**Proposition 1.** *If $M$ is a full rank diagonal matrix, and $u^* \in \mathbb{R}^{n^2 \times n^2}$ satisfies Eq. (7), then $u^*$ satisfies Eq. (4).*

### 2.4.1 JACOBI METHOD

A simple but effective way to choose $M$ is the Jacobi method, which sets $M = I$ (a full rank diagonal matrix, as required by Proposition 1). For Poisson equations, this update rule has the following form,

$$\hat{u}_{i,j} = \frac{1}{4}(u_{i-1,j} + u_{i+1,j} + u_{i,j-1} + u_{i,j+1}) + \frac{h^2}{4}f_{i,j} \tag{8}$$

$$u' = G\hat{u} + (1 - G)b \tag{9}$$

For Poisson equations and any geometry $G$, the update matrix $T = G(I - A)$ has spectral radius $\rho(T) < 1$ (see Appendix B). In addition, by Proposition 1 any fixed point of the update rule Eq.(8,9) must satisfy Eq. (4). Both convergence and fixed point conditions from Definition 1 are satisfied: Jacobi iterator Eq.(8,9) is valid for any Poisson PDE problem.

In addition, each step of the Jacobi update can be implemented as a neural network layer, i.e., Eq. (8) can be efficiently implemented by convolving $u$ with kernel $\begin{pmatrix} 0 & 1/4 & 0 \\ 1/4 & 0 & 1/4 \\ 0 & 1/4 & 0 \end{pmatrix}$ and adding $h^2 f/4$. The "reset" step in Eq. (9) can also be implemented as multiplying $u$ with G and adding the boundary values $(1 - G)b$.

### 2.4.2 MULTIGRID METHOD

The Jacobi method has very slow convergence rate (LeVeque, 2007). This is evident from the update rule, where the value at each grid point is only influenced by its immediate neighbors. To propagate information from one grid point to another, we need as many iterations as their distance on the grid. The key insight of the Multigrid method is to perform Jacobi updates on a downsampled (coarser) grid and then upsample the results. A common structure is the V-cycle (Briggs et al., 2000). In each

V-cycle, there are $k$ downsampling layers followed by $k$ upsampling layers, and multiple Jacobi updates are performed at each resolution. The downsampling and upsampling operations are also called *restriction* and *prolongation*, and are often implemented using weighted restriction and linear interpolation respectively. The advantage of the multigrid method is clear: on a downsampled grid (by a factor of 2) with mesh width $2h$, information propagation is twice as fast, and each iteration requires only 1/4 operations compared to the original grid with mesh width $h$.

## 3 Learning Fast and Provably Correct Iterative PDE Solvers

A PDE problem consists of five components $(\mathcal{A}, \mathcal{G}, \mathcal{f}, \mathcal{b}, n)$. One is often interested in solving the same PDE class $\mathcal{A}$ under varying $\mathcal{f}$, discretization $n$, and boundary conditions $(\mathcal{G}, \mathcal{b})$. For example, solving the Poisson equation under different boundary conditions (e.g., corresponding to different mechanical systems governed by the same physics). In this paper, we fix $\mathcal{A}$ but vary $\mathcal{G}, \mathcal{f}, \mathcal{b}, n$, and learn an iterator that solves a class of PDE problems governed by the same $\mathcal{A}$. For a discretized PDE problem $(A, G, f, b, n)$ and given a standard (hand designed) iterative solver $\Psi$, our goal is to improve upon $\Psi$ and learn a solver $\Phi$ that has (1) correct fixed point and (2) fast convergence (on average) on the class of problems of interest. We will proceed to parameterize a family of $\Phi$ that satisfies (1) by design, and achieve (2) by optimization.

In practice, we can only train $\Phi$ on a small number of problems $(A, f_i, G_i, b_i, n_i)$. To be useful, $\Phi$ must deliver good performance on every choice of $G, f, b$, and different grid sizes $n$. We show, theoretically and empirically, that our iterator family has good generalization properties: even if we train on a single problem $(A, G, f, b, n)$, the iterator performs well on very different choices of $G, f, b$, and grid size $n$. For example, we train our iterator on a $64 \times 64$ square domain, and test on a $256 \times 256$ L-shaped domain (see Figure 1).

### 3.1 Formulation

For a fixed PDE problem class $\mathcal{A}$, let $\Psi$ be a standard linear iterative solver known to be valid. We will use more formal notation $\Psi(u; G, f, b, n)$ as $\Psi$ is a function of $u$, but also depends on $G, f, b, n$. Our assumption is that for any choice of $G, f, b, n$ (but fixed PDE class $\mathcal{A}$), $\Psi(u; G, f, b, n)$ is valid. We previously showed that Jacobi iterator Eq.(8,9) have this property for the Poisson PDE class.

We design our new family of iterators $\Phi_H : \mathbb{R}^{n^2} \to \mathbb{R}^{n^2}$ as

$$
\begin{aligned}
w &= \Psi(u; G, f, b, n) - u \\
\Phi_H(u; G, f, b, n) &= \Psi(u; G, f, b, n) + GHw
\end{aligned}
\tag{10}
$$

where $H$ is a learned linear operator (it satisfies $H0 = 0$). The term $GHw$ can be interpreted as a correction term to $\Psi(u; G, f, b, n)$. When there is no confusion, we neglect the dependence on $G, f, b, n$ and denote as $\Psi(u)$ and $\Phi_H(u)$.

$\Phi_H$ should have similar computation complexity as $\Psi$. Therefore, we choose $H$ to be a convolutional operator, which can be parameterized by a deep linear convolutional network. We will discuss the parameterization of $H$ in detail in Section 3.4; we first prove some parameterization independent properties.

The correct PDE solution is a fixed point of $\Phi_H$ by the following lemma:

**Lemma 1.** *For any PDE problem $(A, G, f, b, n)$ and choice of $H$, if $u^*$ is a fixed point of $\Psi$, it is a fixed point of $\Phi_H$ in Eq. (10).*

*Proof.* Based on the iterative rule in Eq. (10), if $u^*$ satisfies $\Psi(u^*) = u^*$ then $w = \Psi(u^*) - u^* = \mathbf{0}$. Therefore, $\Phi_H(u^*) = \Psi(u^*) + GH\mathbf{0} = u^*$. □

Moreover, the space of $\Phi_H$ subsumes the standard solver $\Psi$. If $H = 0$, then $\Phi_H = \Psi$. Furthermore, denote $\Psi(u) = Tu + c$, then if $H = T$, then since $GT = T$ (see Eq. (7)),

$$
\Phi_H(u) = \Psi(u) + GT(\Psi(u) - u) = T\Psi(u) + c = \Psi^2(u)
\tag{11}
$$

which is equal to two iterations of $\Psi$. Computing $\Psi$ requires one convolution $T$, while computing $\Phi_H$ requires two convolutions: $T$ and $H$. Therefore, if we choose $H = T$, then $\Phi_H$ computes two iterations of $\Psi$ with two convolutions: it is at least as efficient as the standard solver $\Psi$.

## 3.2 TRAINING AND GENERALIZATION

We train our iterator $\Phi_H(u; G, f, b, n)$ to converge quickly to the ground truth solution on a set $\mathcal{D} = \{(G_l, f_l, b_l, n_l)\}_{l=1}^M$ of problem instances. For each instance, the ground truth solution $u^*$ is obtained from the existing solver $\Psi$. The learning objective is then

$$\min_H \sum_{(G_l, f_l, b_l, n_l) \in \mathcal{D}} \mathbb{E}_{u^0 \sim \mathcal{N}(0,1)} \|\Phi_H^k(u^0; G_l, f_l, b_l, n_l) - u^*\|_2^2 \tag{12}$$

Intuitively, we look for a matrix $H$ such that the corresponding iterator $\Phi_H$ will get us as close as possible to the solution in $k$ steps, starting from a random initialization $u^0$ sampled from a white Gaussian. $k$ in our experiments is uniformly chosen from $[1, 20]$, similar to the procedure in (Song et al., 2017). Smaller $k$ is easier to learn with less steps to back-propagate through, while larger $k$ better approximates our test-time setting: we care about the final approximation accuracy after a given number of iteration steps. Combining smaller and larger $k$ performs best in practice.

We show in the following theorem that there is a convex open set of $H$ that the learning algorithm can explore. To simplify the statement of the theorem, for any linear iterator $\Phi(u) = Tu + c$ we will refer to the spectral radius (norm) of $\Phi$ as the spectral radius (norm) of $T$.

**Theorem 2.** *For fixed $G, f, b, n$, the spectral norm of $\Phi_H(u; G, f, b, n)$ is a convex function of $H$, and the set of $H$ such that the spectral norm of $\Phi_H(u; G, f, b, n) < 1$ is a convex open set.*

*Proof.* See Appendix A. □

Therefore, to find an iterator with small spectral norm, the learning algorithm only has to explore a convex open set. Note that Theorem 2 holds for *spectral norm*, whereas validity requires small *spectral radius* in Theorem 1. Nonetheless, several important PDE problems (Poisson, Helmholtz, etc) are symmetric, so it is natural to use a symmetric iterator, which means that spectral norm is equal to spectral radius. In our experiments, we do not explicitly enforce symmetry, but we observe that the optimization finds symmetric iterators automatically.

For training, we use a single grid size $n$, a single geometry $G$, $f = 0$, and a restricted set of boundary conditions $b$. The geometry we use is a square domain shown in Figure 1a. Although we train on a single domain, the model has surprising generalization properties, which we show in the following:

**Proposition 2.** *For fixed $A, G, n$ and fixed $H$, if for some $f_0, b_0$, $\Phi_H(u; G, f_0, b_0, n)$ is valid for the PDE problem $(A, G, f_0, b_0, n)$, then for all $f$ and $b$, the iterator $\Phi_H(u; G, f, b, n)$ is valid for the PDE problem $(A, G, f, b, n)$.*

*Proof.* See Appendix A. □

The proposition states that we freely generalize to different $f$ and $b$. There is no guarantee that we can generalize to different $G$ and $n$. Generalization to different $G$ and $n$ has to be empirically verified: in our experiments, our learned iterator converges to the correct solution for a variety of grid sizes $n$ and geometries $G$, even though it was only trained on one grid size and geometry.

Even when generalization fails, there is no risk of obtaining incorrect results. The iterator will simply fail to converge. This is because according to Lemma 1, fixed points of our new iterator is the same as the fixed point of hand designed iterator $\Psi$. Therefore if our iterator is convergent, it is valid.

## 3.3 INTERPRETATION OF $H$

What is $H$ trying to approximate? In this section we show that we are training our linear function $GH$ to approximate $T(I - T)^{-1}$: if it were able to approximate $T(I - T)^{-1}$ perfectly, our iterator $\Phi_H$ will converge to the correct solution in a single iteration.

Let the original update rule be $\Psi(u) = Tu + c$, and the unknown ground truth solution be $u^*$ satisfying $u^* = Tu^* + c$. Let $r = u^* - u$ be the current error, and $e = u^* - \Psi(u)$ be the new error after applying one step of $\Psi$. They are related by

$$e = u^* - \Psi(u) = u^* - (Tu + c) = T(u^* - u) = Tr \tag{13}$$

In addition, let $w = \Psi(u) - u$ be the update $\Psi$ makes. This is related to the current error $r$ by

$$w = \Psi(u) - u = Tu + c - u + (u^* - Tu^* - c) = T(u - u^*) + (u^* - u) = (I - T)r \quad (14)$$

From Eq. (10) we can observe that the linear operator $GH$ takes as input $\Psi$'s update $w$, and tries to approximate the error $e$: $GHw \approx e$. If the approximation were perfect: $GHw = e$, the iterator $\Phi_H$ would converge in a single iteration. Therefore, we are trying to find some linear operator $R$, such that $Rw = e$. In fact, if we combine Eq. (13) and Eq. (14), we can observe that $T(I - T)^{-1}$ is (uniquely) the linear operator we are looking for

$$T(I - T)^{-1}w = e \quad (15)$$

where $(I - T)^{-1}$ exists because $\rho(T) < 1$, so all eigenvalues of $I - T$ must be strictly positive. Therefore, we would like our linear function $GH$ to approximate $T(I - T)^{-1}$.

Note that $(I - T)^{-1}$ is a dense matrix in general, meaning that it is impossible to exactly achieve $GH = T(I - T)^{-1}$ with a convolutional operator $H$. However, the better $GH$ is able to approximate $T(I - T)^{-1}$, the faster our iterator converges to the solution $u^*$.

### 3.4 LINEAR DEEP NETWORKS

In our iterator design, $H$ is a linear function parameterized by a linear deep network without non-linearity or bias terms. Even though our objective in Eq. (12) is a non-linear function of the parameters of the deep network, this is not an issue in practice. In particular, Arora et al. (2018) observes that when modeling linear functions, deep networks can be faster to optimize with gradient descent compared to linear ones, despite non-convexity.

Even though a linear deep network can only represent a linear function, it has several advantages. On an $n \times n$ grid, each convolution layer only requires $O(n^2)$ computation and have a constant number of parameters, while a general linear function requires $O(n^4)$ computation and have $O(n^4)$ parameters. Stacking $d$ convolution layers allows us to parameterize complex linear functions with large receptive fields, while only requiring $O(dn^2)$ computation and $O(d)$ parameters. We experiment on two types of linear deep networks:

**Conv model.** We model $H$ as a network with $3 \times 3$ convolutional layers without non-linearity or bias. We will refer to a model with $k$ layers as "Conv$k$", e.g. Conv3 has 3 convolutional layers.

**U-Net model.** The Conv models suffer from the same problem as Jacobi: the receptive field grows only by 1 for each additional layer. To resolve this problem, we design the deep network counterpart of the Multigrid method. Instead of manually designing the sub-sampling / super-sampling functions, we use a U-Net architecture (Ronneberger et al., 2015) to learn them from data. Because each layer reduces the grid size by half, and the $i$-th layer of the U-Net only operates on $(2^{-i}n)$-sized grids, the total computation is only increased by a factor of

$$1 + 1/4 + 1/16 + \cdots < 4/3$$

compared to a two-layer convolution. The minimal overhead provides a very large improvement of convergence speed in our experiments. We will refer to Multigrid and U-Net models with $k$ sub-sampling layers as Multigrid$k$ and U-Net$k$, e.g. U-Net2 is a model with 2 sub-sampling layers.

## 4 EXPERIMENTS

### 4.1 SETTING

We evaluate our method on the 2D Poisson equation with Dirichlet boundary conditions, $\nabla^2 u = f$. There exist several iterative solvers for the Poisson equation, including Jacobi, Gauss-Seidel, conjugate-gradient, and multigrid methods. We select the Jacobi method as our standard solver $\Psi$.

To reemphasize, our goal is to train a model on simple domains where the ground truth solutions can be easily obtained, and then evaluate its performance on different geometries and boundary conditions. Therefore, for training, we select the simplest Laplace equation, $\nabla^2 u = 0$, on a square domain with boundary conditions such that each side is a random fixed value. Figure 1a shows an

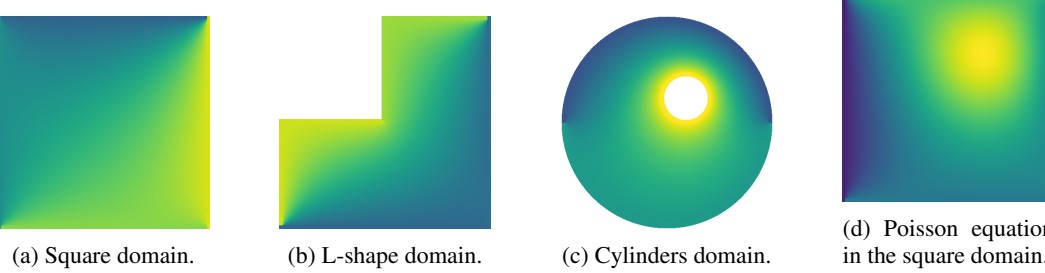

(a) Square domain.   (b) L-shape domain.   (c) Cylinders domain.   (d) Poisson equation in the square domain.

Figure 1: The ground truth solutions of examples in different settings. We only train our models on the square domain, and we test on all 4 settings.

example of our training domain and its ground truth solution. This setting is also used in Farimani et al. (2017) and Sharma et al. (2018).

For testing, we use larger grid sizes than training. For example, we test on $256 \times 256$ grid for a model trained on $64 \times 64$ grids. Moreover, we designed challenging geometries to test the generalization of our models. We test generalization on 4 different settings: (i) same geometry but larger grid, (ii) L-shape geometry, (iii) Cylinders geometry, and (iv) Poisson equation in same geometry, but $f \neq 0$. The two geometries are designed because the models were trained on square domains and have never seen sharp or curved boundaries. Examples of the 4 settings are shown in Figure 1.

### 4.2 EVALUATION

As discussed in Section 2.4, the convergence rate of any linear iterator can be determined from the spectral radius $\rho(T)$, which provides guarantees on convergence and convergence rate. However, a fair comparison should also consider the computation cost of $H$. Thus, we evaluate the convergence rate by calculating the computation cost required for the error to drop below a certain threshold.

On GPU, the Jacobi iterator and our model can both be efficiently implemented as convolutional layers. Thus, we measure the computation cost by the number of convolutional layers. On CPU, each Jacobi iteration $u'_{i,j} = \frac{1}{4}(u_{i-1,j} + u_{i+1,j} + u_{i,j-1} + u_{i,j+1})$ has 4 multiply-add operations, while a $3 \times 3$ convolutional kernel requires 9 operations, so we measure the computation cost by the number of multiply-add operations. This metric is biased in favor of Jacobi because there is little practical reason to implement convolutions on CPU. Nonetheless, we report both metrics in our experiments.

### 4.3 CONV MODEL

Table 1 shows results of the Conv model. The model is trained on a $16 \times 16$ square domain, and tested on $64 \times 64$. For all settings, our models converge to the correct solution, and require less computation than Jacobi. The best model, Conv3, is $\sim 5\times$ faster than Jacobi in terms of layers, and $\sim 2.5\times$ faster in terms of multiply-add operations.

As discussed in Section 3.2, if our iterator converges for a geometry, then it is guaranteed to converge to the correct solution for any $f$ and boundary values $b$. The experiment results show that our model not only converges but also converges faster than the standard solver, even though it is only trained on a smaller square domain.

### 4.4 U-NET MODEL

For the U-Net models, we compare them against Multigrid models with the same number of subsampling and smoothing layers. Therefore, our models have the same number of convolutional layers, and roughly $9/4$ times the number of operations compared to Multigrid. The model is trained on a $64 \times 64$ square domain, and tested on $256 \times 256$.

The bottom part of Table 1 shows the results of the U-Net model. Similar to the results of Conv models, our models outperforms Multigrid in all settings. Note that U-Net2 has lower computation

Table 1: Comparisons between our models and the baseline solvers. The Conv models are compared with Jacobi, and the U-Net models are compared with Multigrid. The numbers are the ratio between the computation costs of our models and the baselines. None of the values are greater than 1, which means that all of our models achieve a speed up on every problem and both performance metric (convolutional layers and multiply-add operations).

| Model | Baseline | Square layers / ops | L-shape layers / ops | Cylinders layers / ops | Square-Poisson layers / ops |
|-------|----------|---------------------|----------------------|------------------------|------------------------------|
| Conv1 | Jacobi | 0.432 / 0.702 | 0.432 / 0.702 | 0.432 / 0.702 | 0.431 / 0.701 |
| Conv2 | Jacobi | 0.286 / 0.524 | 0.286 / 0.524 | 0.286 / 0.524 | 0.285 / 0.522 |
| Conv3 | Jacobi | **0.219 / 0.424** | **0.219 / 0.423** | **0.220 / 0.426** | **0.217 / 0.421** |
| Conv4 | Jacobi | 0.224 / 0.449 | 0.224 / 0.449 | 0.224 / 0.448 | 0.222 / 0.444 |
| U-Net2 | Multigrid2 | **0.091 / 0.205** | **0.090 / 0.203** | **0.091 / 0.204** | **0.079 / 0.178** |
| U-Net3 | Multigrid3 | 0.220 / 0.494 | 0.213 / 0.479 | 0.201 / 0.453 | 0.185 / 0.417 |

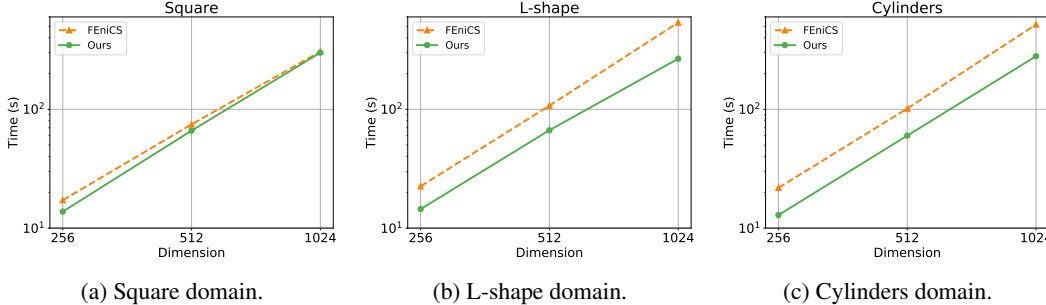

(a) Square domain.   (b) L-shape domain.   (c) Cylinders domain.

Figure 2: CPU runtime comparisons of our model with the FEniCS model. Our method is comparable or faster than the best solver in FEniCS in all cases. When run on GPU, our solver provides an additional 30× speedup.

cost compared with Multigrid2 than U-Net3 compared to Multigrid 3. This is because Multigrid2 is a relatively worse baseline. U-Net3 still converges faster than U-Net2.

## 4.5 Comparison with FEniCS

The FEniCS package (Logg et al., 2012) provides a collection of tools with high-level Python and C++ interfaces to solve differential equations. The open-source project is developed and maintained by a global community of scientists and software developers. Its extensive optimization over the years, including the support for parallel computation, has led to its widespread adaption in industry and academia (Alnæs et al., 2015).

We measure the wall clock time of the FEniCS model and our model, run on the same hardware. The FEniCS model is set to be the minimal residual method with algebraic multigrid preconditioner, which we measure to be the fastest compared to other methods such as Jacobi or Incomplete LU factorization preconditioner. We ignore the time it takes to set up geometry and boundary conditions, and only consider the time the solver takes to solve the problem. We set the error threshold to be 1 percent of the initial error. For the square domain, we use a quadrilateral mesh. For the L-shape and cylinder domains, however, we let FEniCS generate the mesh automatically, while ensuring the number of mesh points to be similar.

Figure 2 shows that our model is comparable or faster than FEniCS in wall clock time. These experiments are all done on CPU. Our model efficiently runs on GPU, while the fast but complex methods in FEniCS do not have efficient GPU implementations available. On GPU, we measure an additional 30× speedup (on Tesla K80 GPU, compared with a 64-core CPU).

## 5 RELATED WORK

Recently, there have been several works on applying deep learning to solve the Poisson equation. However, to the best of our knowledge, previous works used deep networks to directly generate the solution; they have no correctness guarantees and are not generalizable to arbitrary grid sizes and boundary conditions. Most related to our work are (Farimani et al., 2017) and (Sharma et al., 2018), which learn deep networks to output the solution of the 2D Laplace equation (a special case where $f = 0$). (Farimani et al., 2017) trained a U-Net model that takes in the boundary condition as a 2D image and outputs the solution. The model is trained by L1 loss to the ground truth solution and an adversarial discriminator loss. (Sharma et al., 2018) also trained a U-net model but used a weakly-supervised loss. There are other related works that solved the Poisson equation in concrete physical problems. (Tang et al., 2017) solved for electric potential in 2D/3D space; (Tompson et al., 2017) solved for pressure fields for fluid simulation; (Zhang et al., 2018) solved particle simulation of a PN Junction.

There are other works that solve other types of PDEs. For example, many studies aimed to use deep learning to accelerate and approximate fluid dynamics, governed by the Euler equation or the Navier-Stokes equations (Guo et al., 2016; Yang et al., 2016; Chu & Thuerey, 2017; Kutz, 2017). (Eismann et al., 2018) use Bayesian optimization to design shapes with reduced drag coefficients in laminar fluid flow. Other applications include solving the Schrodinger equation (Mills et al., 2017), turbulence modeling (Singh et al., 2017), and the American options and Black Scholes PDE (Sirignano & Spiliopoulos, 2018). A lot of these PDEs are nonlinear and may not have a standard linear iterative solver, which is a limitation to our current method since our model must be built on top of an existing linear solver to ensure correctness. We consider the extension to different PDEs as future work.

## 6 CONCLUSION

We presented a method to learn an iterative solver for PDEs that improves on an existing standard solver. The correct solution is theoretically guaranteed to be the fixed point of our iterator. We show that our model, trained on simple domains, can generalize to different grid sizes, geometries and boundary conditions. It converges correctly and achieves significant speedups compared to standard solvers, including highly optimized ones implemented in FEniCS.

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

## A  PROOFS

**Theorem 1.** *For a linear iterator $\Psi(u) = Tu + c$, $\Psi$ converges to a unique stable fixed point from any initialization if and only if the spectral radius $\rho(T) < 1$.*

*Proof.* Suppose $\rho(T) < 1$, then $(I-T)^{-1}$ must exist because all eigenvalues of $I-T$ must be strictly positive. Let $u^* = (I - T)^{-1}c$; this $u^*$ is a stationary point of the iterator $\Psi$, i.e. $u^* = Tu^* + c$. For any initialization $u^0$, let $u^k = \Psi^k(u^0)$. The error $e^k = u^* - u^k$ satisfies

$$Te^k = (Tu^* + c) - (Tu^k + c) = u^* - u^{k+1} = e^{k+1} \Rightarrow e^k = T^k e^0 \tag{16}$$

Since $\rho(T) < 1$, we know $T^k \to 0$ as $k \to \infty$ (LeVeque, 2007), which means the error $e^k \to 0$. Therefore, $\Psi$ converges to $u^*$ from any $u^0$.

Now suppose $\rho(T) \geq 1$. Let $\lambda_1$ be the largest absolute eigenvalue where $\rho(T) = |\lambda_1| \geq 1$, and $v_1$ be its corresponding eigenvector. We select initialization $u^0 = u^* + v_1$, then $e^0 = v_1$. Because $|\lambda_1| \geq 1$, we have $|\lambda_1^k| \geq 1$, then

$$T^k e^0 = \lambda_1^k v_1 \not\to_{k \to \infty} 0$$

However we know that under a different initialization $\hat{u}^0 = u^*$, we have $\hat{e}^0 = 0$, so $T^k \hat{e}^0 = 0$. Therefore the iteration cannot converge to the same fixed point from different initializations $u^0$ and $\hat{u}^0$.

$\square$

**Proposition 1** *If $M$ is a full rank diagonal matrix, and $u^* \in \mathbb{R}^{n^2 \times n^2}$ satisfies Eq. (7), then $u^*$ satisfies Eq. (4).*

*Proof of Proposition 1.* Let $u^*$ be a fixed point of Eq. (7) then

$$Gu^* + (I - G)u^* = G(M^{-1}Nu^* + M^{-1}f) + (I - G)b$$

This is equivalent to

$$(I - G)u^* = (I - G)b$$
$$G(u^* - M^{-1}Nu^* - M^{-1}f) = 0 \tag{17}$$

The latter equation is equivalent to $GM^{-1}(Au^* - f) = 0$. If $M$ is a full rank diagonal matrix, this implies $G(Au^* - f) = 0$, which is $GAu^* = Gf$. Therefore, $u^*$ satisfies Eq.(4). $\square$

**Theorem 2.** *For fixed $G, f, b, n$, the spectral norm of $\Phi_H(u; G, f, b, n)$ is a convex function of $H$, and the set of $H$ such that the spectral norm of $\Phi_H(u; G, f, b, n) < 1$ is a convex open set.*

*Proof.* As before, denote $\Psi(u) = Tu + c$. Observe that

$$\Phi_H(u; G, f, b, n) = Tu + c + GH(Tu + c - u) = (T + GHT - GH)u + GHc + c \tag{18}$$

The spectral norm $\|\cdot\|_2$ is convex with respect to its argument, and $(T + GHT - GH)$ is linear in $H$. Thus, $\|T + GHT - GH\|_2$ is convex in $H$ as well. Thus, under the condition that $\|T + GHT - GH\|_2 < 1$, the set of $H$ must be convex because it is a sub-level set of the convex function $\|T + GHT - GH\|_2$.

To prove that it is open, observe that $\|\cdot\|_2$ is a continuous function, so $\|T + GHT - GH\|_2$ is a continuous map from $H$ to the spectral radius of $\Phi_H$. If we consider the set of $H$ such that $\|T + GHT - GH\|_2 < 1$, this set is the preimage of $(-\epsilon, 1)$ for any $\epsilon > 0$. As $(-\epsilon, 1)$ is open, its preimage must be open.

$\square$

**Proposition 2.** *For fixed $A, G, n$ and fixed $H$, if for some $f_0, b_0$, $\Phi_H(u; G, f_0, b_0, n)$ is valid for the PDE problem $(A, G, f_0, b_0, n)$, then for all $f$ and $b$, the iterator $\Phi_H(u; G, f, b, n)$ is valid for the PDE problem $(A, G, f, b, n)$.*

*Proof.* From Theorem 1 and Lemma 1, our iterator is valid if and only if $\rho(T + GHT - GH) < 1$. The iterator $T + GHT - GH$ only depends on $A, G$, and is independent of the constant $c$ in Eq. (18). Thus, the validity of the iterator is independent with $f$ and $b$. Thus, if the iterator is valid for some $f_0$ and $b_0$, then it is valid for any choice of $f$ and $b$.

$\square$

## B   PROOF OF CONVERGENCE OF JACOBI METHOD

In Section 2.4.1, we show that for Poisson equation, the update matrix $T = G(I - A)$. We now formally prove that $\rho(G(I - A)) < 1$ for any $G$.

For any matrix $T$, the spectral radius is bounded by the spectral norm: $\rho(T) \leq \|T\|_2$, and the equality holds if $T$ is symmetric. Since $(I - A)$ is a symmetric matrix, $\rho(I - A) = \|I - A\|_2$. It has been proven that $\rho(I - A) < 1$ (Frankel, 1950). Moreover, $\|G\|_2 = 1$. Finally, matrix norms are sub-multiplicative, so

$$\rho(T) \leq \|G(I - A)\|_2 \leq \|G\|_2 \|I - A\|_2 < 1 \tag{19}$$

$\rho(T) < 1$ is true for any $G$. Thus, the standard Jacobi method is valid for the Poisson equation under any geometry.

