# OpenReview forum: "Learning Neural PDE Solvers with Convergence Guarantees"
_ICLR.cc/2019/Conference_

### Official Review · AnonReviewer3 · 2018-10-29
**A linear method for speeding up PDE solvers with good empirical performances**

**Rating:** 6
**Confidence:** 3

**Review:**

Summary:
The authors propose a method to learn and improve problem-tailored PDE solvers from existing ones. The linear updates of the target solver, specified by the problem's geometry and boundary conditions, are computed from the updates of a well-known solver through an optimized linear map.  The obtained solver is guaranteed to converge to the correct solution and
achieves a considerable speed-up compared to solvers obtained from alternative state-of-the-art methods.

Strengths:
Solving PDEs is an important and hard problem and the proposed method seems to consistently outperform the state of the art. I ve liked the idea of learning a speed-up operator to improve the performance of a standard solver and adapt it to new boundary conditions or problem geometries. The approach is simple enough to allow a straightforward proof of correctness.

Weaknesses:
The method seems to rely strongly on the linearity of the solver and its deformation (to guarantee the correctness of the solution). The operator H is a matrix of finite dimensions and it is not completely clear to me what is the role of the multi-layer parameterization. Based on a grid approach, the idea applies only to one- or two-dimensional problems.

Questions:
- in the introduction, what does it mean that generic solvers are effective 'but could be far from optimal'?  Does this refer to the convergence speed or to the correctness of the solution?
- other deep learning approaches to PDE solving are mentioned in the introduction. Is the proposed method compared to them somewhere in the experiments?
- given a PDE and some boundary conditions, is there any known method to choose the liner iterator T optimally? For example, since u* is the solution of a linear system, could one choose the updates to be the gradient descent updates of a least-squares objective such as || A u - f||^2?
- why is the deep network parameterization needed? Since no nonlinearities are present, isn t this equivalent to fix the rank of H?
- given the `  interpretation of H' sketched in Section 3.3, is there any relationship between the proposed accelerated update and the update of second-order coordinated descent methods (like Newton or quasi-Newton)?

---

> ### Author Response · Authors · 2018-11-16
> **Reply to AnonReviewer3**
>
> Thank you for your helpful reviews and suggestions.
>
> 1) “The method seems to rely strongly on the linearity of the solver and its deformation (to guarantee the correctness of the solution). The operator H is a matrix of finite dimensions and it is not completely clear to me what is the role of the multi-layer parameterization.
> “why is the deep network parameterization needed? Since no nonlinearities are present, isn t this equivalent to fix the rank of H?”
>
> Even though composition of linear functions is still linear, using d linear layers is better than one. On a grid with n^2 vertices: one convolution layer requires O(n^2) computations and have local receptive field; one fully-connected layer requires O(n^4) computations and have global receptive field; our deep U-Net architecture has O(n^2) computations but global receptive field. Our hope is that the deep U-Net architecture learns a linear function with both good computation properties (O(n^2)) and convergence properties, which is impossible for one layer models.
>
> Our learned network H is a convolutional operator, which does not have low rank. A low rank H is unlikely to perform well because many different errors may be mapped to the same correction term, while a high rank H can correct different errors differently. Our parameterization learns a high rank H with O(n^2) computation.
>
>
> 2) “Based on a grid approach, the idea applies only to one- or two-dimensional problems.”
>
> Our method generalizes without modification to any dimensional problems: simply replace 2-D convolution to k-D convolution.
>
>
> 3) “in the introduction, what does it mean that generic solvers are effective 'but could be far from optimal'?  Does this refer to the convergence speed or to the correctness of the solution?”
>
> We meant that generic solvers like Jacobi are hand-designed and theoretically correct, but may not be optimal in terms of convergence speed. Designing a solver is a trade-off between computation-per-iteration and spectral radius. We would like to have the smallest spectral radius given computation budget. We verify in our experiments: human designed solvers (e.g. Jacobi) are not Pareto optimal, and are outperformed by our learned solvers. Similar observations have also been made in other fields: learned models outperform hand designed ones, e.g. Andrychowicz et al., 2016, Song et al, 2017.
>
>
> 4) “other deep learning approaches to PDE solving are mentioned in the introduction. Is the proposed method compared to them somewhere in the experiments?”
>
> To the best of our knowledge, related works applying ML to PDEs directly fit the solution with deep networks, which have no correctness or generalization guarantees and are restricted to specific dimensions and geometries. Our algorithm is the first deep learning based method with provable correctness and generalization guarantees.
>
>
> 5) “given a PDE and some boundary conditions, is there any known method to choose the liner iterator T optimally? For example, since u* is the solution of a linear system, could one choose the updates to be the gradient descent updates of a least-squares objective such as || A u - f||^2?”
>
> Actually, this is exactly the update rule for most existing methods (conjugate gradient, Jacobi, etc). We compared with these methods (conjugate gradient, Jacobi) in experiments and outperform them. For example, if we minimize the objective 1/2 u^T A u - u^T f, given that A is symmetric, positive-definite, this objective has a unique minimizer, and the derivative is exactly Au - f. If we perform gradient descent on this objective with learning rate 1, we get exactly the Jacobi update.
>
> Gradient descent may not be optimal; improving it is undergoing active research (e.g. ADAM, Adagrad, “Learning to learn” [1]). We tackle a special class of optimization problems and design methods with both correctness guarantees and better performance.
>
> [1] Andrychowicz, Marcin, et al. "Learning to learn by gradient descent by gradient descent. NIPS, 2016.
>
>
> 6) “given the `interpretation of H' sketched in Section 3.3, is there any relationship between the proposed accelerated update and the update of second-order coordinated descent methods (like Newton or quasi-Newton)?”
>
> On a grid with k (= n^2) vertices, second order methods (e.g. Newton) have optimal convergence speed (solve linear equations with a single update), but poor computation complexity (O(k^3) to solve A inverse). Our method only require O(k) computation per-iteration, and we hope to achieve a good trade-off between convergence speed and computation budget by optimization.

---

### Official Review · AnonReviewer1 · 2018-10-30
**Interesting, well-written paper**

**Rating:** 8
**Confidence:** 4

**Review:**

==Summary==
This paper is well-executed and interesting. It does a good job of bridging the gap between distinct bodies of literature, and is very in touch with modern ML ideas.

I like this paper and advocate that it is accepted. However, I expect that it would have higher impact if it appeared in the numerical PDE community. I encourage you to consider this conference paper to be an early version of a more comprehensive piece of work to be released to that community.

My main critique is that the paper needs to do a better job of discussing prior work on data-driven methods for improving PDE solvers.
==Major comments==
* You need to spend considerably more space discussing the related work on using ML to improve PDE solvers. Most readers will be unfamiliar with this. You should explain what they do and how they are qualitatively different than your approach.

* You do a good job 3.3 of motivating for what H is doing. However, you could do a better job of motivating the overall setup of (6). Is this a common formulation? If so, where else is it used?
* I’m surprised that you didn’t impose some sort of symmetry conditions on the convolutions in H, such as that they are invariant to flips of the kernel. This is true, for example, for the linearized Poisson operator.

==Minor comments==

* Valid iterators converge to a valid solution. However, can’t there be multiple candidate solutions? How would you construct a method that would be able to find all possible solutions?

* In (9), why do you randomize the value of k? Wouldn’t you want to learn a different H depending on what computation budget you knew you were going to use downstream when you deploy the solver?

* In future work it may make sense to learn a different H_i for each step i of the iterative solver.

* When introducing iterative solvers, you leave it as an afterthought that b will be enforced by clamping values at the end of each iteration. This seems like a pretty important design decision. Are there alternatives that guarantee that u satisfies b always, rather than updating u in such a way that it violates G and then clamping it back? Along these lines, it might be useful to pose (2) with additional terms in the linear system to reflect G.

---

> ### Author Response · Authors · 2018-11-15
> **Reply to AnonReviewer1**
>
> Thank you for your helpful reviews and suggestions.
>
> 1) “You need to spend considerably more space discussing the related work on using ML to improve PDE solvers. Most readers will be unfamiliar with this. You should explain what they do and how they are qualitatively different than your approach.”
>
> We will add more discussions in our updated paper. To the best of our knowledge, related works applying ML to PDEs directly fit the solution with deep networks, which have no correctness or generalization guarantees and are restricted to specific dimensions and geometries.
>
>
> 2) “You do a good job 3.3 of motivating for what H is doing. However, you could do a better job of motivating the overall setup of (6). Is this a common formulation? If so, where else is it used?”
>
> This formulation is a novel idea that provides correctness guarantees by leveraging a hand designed solver: we modify the residual of a hand designed solver. Another idea that also modify the residual (but not of a hand designed solver) is conjugate gradient.
>
>
> 3) “I’m surprised that you didn’t impose some sort of symmetry conditions on the convolutions in H, such as that they are invariant to flips of the kernel. This is true, for example, for the linearized Poisson operator.”
>
> Generalization, for our model, is almost for free because of our linear ConvNet setup. Therefore, we didn’t find strong reasons to restrict the network parameters and reduce dimensionality. Enforcing symmetry introduces unnecessary overhead.
>
>
> 4) “Valid iterators converge to a valid solution. However, can’t there be multiple candidate solutions? How would you construct a method that would be able to find all possible solutions?”
>
> For most PDEs with Dirichlet boundary conditions (e.g. Possion, Helmholtz), the solution is always unique. Thus, a valid iterator should converge to the unique solution. We currently consider PDEs that have unique solutions.
>
>
> ==Minor comments==
>
> 5) “In (9), why do you randomize the value of k? Wouldn’t you want to learn a different H depending on what computation budget you knew you were going to use downstream when you deploy the solver?”
>
> Our hope is to learn a generic solver for a type of PDE that can be applied to a variety of applications. Therefore, we train the model agnostic to downstream applications. Nonetheless practitioners who know their computation budget can certainly fine tune our iterator with a fixed k.
>
> 6) “In future work it may make sense to learn a different H_i for each step i of the iterative solver.”
>
> Thank you for the suggestion. We can try in the future, e.g. there are some methods that take the history of the iteration into account, which means it has a different H for each step.
>
>
> 7) “When introducing iterative solvers, you leave it as an afterthought that b will be enforced by clamping values at the end of each iteration. This seems like a pretty important design decision. Are there alternatives that guarantee that u satisfies b always, rather than updating u in such a way that it violates G and then clamping it back? Along these lines, it might be useful to pose (2) with additional terms in the linear system to reflect G.”
>
> This is the most straightforward way to satisfy the boundary condition, and most existing iterative solvers enforce boundary conditions with this reset operation. We will also explicitly add G into our update rule in our updated paper.

---

### Official Review · AnonReviewer2 · 2018-11-02
**A Good and Solid Work**

**Rating:** 7
**Confidence:** 4

**Review:**

This paper develops a method to accelerate the finite difference method in solving PDEs. Basically, the paper proposes a revised framework for fixed point iteration after discretization. The framework introduces a free linear operator --- the choice of the linear operator will influence the convergence rate. The paper uses a deep linear neural network to learn a good operator. Experimental results on Poisson equations show that the learned operator achieves significant speed-ups. The paper also gives theoretical analysis about the range of the valid linear operator (convex open set) and guarantees of the generalization for the learned operator.

This is, in general, a good paper. The work is solid and results promising.  Solving PDEs is no doubt an important problem, having broad applications. It will be very meaningful if we can achieve the same accuracy using much less computational power.  Here, I have a few questions.

1). Why didn’t you try the nonlinear deep network? Is it merely for computational efficiency? I expect that nonlinear networks might result in even better estimates of H and further reduce the number of fixed-point iterations, despite each operation of H will be more expensive. There might be some trade-off here. But I would like to see some empirical results and discussions.

2). The evaluation is only on Poisson equations, which are known to be easy. Have you tried other PDEs, such as Burger’s equations? I think your method will be more meaningful for those challenging PDEs, because they will require much more fine-grained grids to achieve a satisfactory accuracy and hence much more expensive. It will be great if your method can dramatically improve the efficiency for solving these equations.

3). I am a bit confused about the statement of Th 3 --- the last sentence “H is valid for all parameters f and b if the iterator \psi converges …” I think it should be “for one parameter”.

Miscellaneous:
1)	Typo. In eq. (7)
2)	Section 3.3, H(w) should be Hw (for consistency)

---

> ### Author Response · Authors · 2018-11-15
> **Reply to AnonReviewer2**
>
> Thank you for your helpful reviews and suggestions.
>
> 1) "Why didn’t you try the nonlinear deep network? Is it merely for computational efficiency? I expect that nonlinear networks might result in even better estimates of H and further reduce the number of fixed-point iterations, despite each operation of H will be more expensive. There might be some trade-off here. But I would like to see some empirical results and discussions."
>
> The reason we did not use nonlinear deep networks is that it’s hard to prove correctness guarantees. Our linear iterator has provably correct fixed point while nonlinear iterators may have non-unique or incorrect fixed points. In addition, it is easy to prove convergence by spectral theory, while this is not the case for nonlinear operators.
>
>
> 2). "The evaluation is only on Poisson equations, which are known to be easy. Have you tried other PDEs, such as Burger’s equations? I think your method will be more meaningful for those challenging PDEs, because they will require much more fine-grained grids to achieve a satisfactory accuracy and hence much more expensive. It will be great if your method can dramatically improve the efficiency for solving these equations."
>
> We did additional experiments on Helmholtz equations, \nabla^2 u + k^2 u = 0, which is known to be very challenging [1]. So far we have some preliminary results of Conv1 model in a square domain: we outperform traditional methods by a similar margin. The following show for different values of k, the ratio of computation cost compared to Jacobi in terms of layers / flops (same as Table 1).
> k = 1: 0.422 / 0.685
> k = 2: 0.396 / 0.643
> k = 3: 0.383 / 0.622
>
> We leave more thorough analysis of the Helmholtz equation for future work.
>
> [1] Oliver G. Ernst and Martin J. Gander. Why it is Difficult to Solve Helmholtz Problems with Classical Iterative Methods. Numerical analysis of multiscale problems, 2012.
>
>
> 3). "I am a bit confused about the statement of Th 3 --- the last sentence H is valid for all parameters f and b if the iterator \psi converges … I think it should be 'for one parameter'. "
>
> In Theorem 1 and Lemma 1, we showed that if our iterator is convergent, it converges to the correct solution, hence it is valid. In Theorem 3, we showed that if the iterator is valid for some f and b, then the iterator is valid for every f and b. These combined implies that the iterator is valid for every f and b if it is convergent for one f and b. We will rephrase Theorem 3 to remove the confusion.

---

### Public Comment · (anonymous) · 2018-10-15
**Deep Multigrid**

Hello!
The similar idea is proposed in "Deep Multigrid: learning prolongation and restriction matrices" (https://arxiv.org/abs/1711.03825), where authors optimize parameters of the multigrid method with a neural network reformulation of the multigrid method and automatic differentiation tool. Also, almost the same objective function to measure parameters quality is used, but with explanation how does this objective relate to the spectral radius of the iteration matrix.

---

> ### Author Response · Authors · 2018-10-17
> **Differences and problems with Deep Multigrid**
>
> Hello!
>
> Thank you for pointing out this unpublished but relevant work! We were not aware of it and we will certainly add a reference. Deep Multigrid has some surface resemblance to our method, but there are major differences:
>
> (1) Generalization: Deep Multigrid does not generalize to different grid size or different geometries. The learned prolongation and restriction operators are fully-connected layers, which need retraining for each grid size and geometry. Our model generalizes (both by design and in experiments) to very different grid sizes and geometries after training on a single example (Figure 1).
>
> (2) Usability: Deep Multigrid only experimented on 1D grids, with no proposed generalization to 2D or 3D geometries.
> On 1D grid, the matrix A is tridiagonal, and Au = f can be solved exactly by Gaussian elimination in O(n) time [1]. Contrastly, our method applies without modification to any dimension (by using d-dimensional convolution).
>
> (3) Flexibility: Deep Multigrid only learns prolongation and restriction operators. Our U-Net model is end-to-end: it implicitly includes smoothing, prolongation, and restriction. Our approach is simpler yet more general.
>
> (4) Experiments: Deep multigrid does not compare runtime with state-of-the-art solvers. Our method is faster (wall-clock time and number of operations) than both Jacobi Multigrid and FEniCS.
>
>
> [1] Randall J LeVeque. Finite difference methods for ordinary and partial differential equations: steady-state and time-dependent problems, volume 98. Siam, 2007.

---

### Public Comment · ~francesco_bardi1 · 2019-01-04
**ICLR Reproducibility Challenge 2019 - Team Name: zoidberg**

We participated in ICLR Reproducibility Challenge 2019 and you can find our full report and code, written in Python using PyTorch, on Github: https://github.com/francescobardi/pde_solver_deep_learned.

This paper was very interesting and challenging, it introduced - for us at least - a novel idea on how to apply ML-techniques.

We could partially confirm the results reported in the original paper, not every result was reproducible either through lack of time or certainty in how these results where achieved or measured. We did not have the opportunity to test the solver using the MultiGrid method, nor the square-Poisson problem.

We have some questions regarding different aspects of the paper.

1 Code
- We have not found any reference to the source code used to produce the results, if it is publicly available you could add a reference in the paper.

2 Reset operator
- It is not clear if the proposed approach to enforce boundary condition can be extended to boundary conditions other than Dirichlet or to other iterative methods such as Gauss Seidel.

3 Training process
- How many problem instances were considered to build the loss function?
- What is the range for the random value that is applied to each edge of the square?
- How do you obtain the ground truth solution?
- Are the weights of the convolutional kernels randomly initialized or do you set them?
- What optimization algorithm do you use if any?
- How do you propose to include the spectral radius constraint?

4 Model testing:
- How were the numbers reported in Table 1 obtained?
- How do you implement the cylindrical geometry in a finite difference framework? Did you use radial coordinates or a non-uniform grid?

---

### Meta-Review · Area_Chair1 · 2018-12-10
**A nice example of allowing learning without losing guarantees**

**Confidence:** 3
**Recommendation:** Accept (Poster)

**Metareview:**

Quality: The overall quality of the work is high.  The main idea and technical choices are well-motivated, and the method is about as simple as it could be while achieving its stated objectives.

Clarity:  The writing is clear, with the exception of using alternative scripts for some letters in definitions.

Originality:  The biggest weakness of this work is originality, in that there is a lot of closely related work, and similar ideas without convergence guarantees have begun to be explored.  For example, the (very natural) U-net architecture was explored in previous work.

Significance:  This seems like an example of work that will be of interest both to the machine learning community, and also the numerics community, because it also achieves the properties that the numerics community has historically cared about.  It is significant on its own as an improved method, but also as a demonstration that using deep learning doesn't require scrapping existing frameworks but can instead augment them.